# Adaptive Evolution Compensated for the Plasmid Fitness Costs Brought by Specific Genetic Conflicts

**DOI:** 10.3390/pathogens12010137

**Published:** 2023-01-13

**Authors:** Feifeng Li, Jiong Wang, Ying Jiang, Yingyi Guo, Ningjing Liu, Shunian Xiao, Likang Yao, Jiahui Li, Chuyue Zhuo, Nanhao He, Baomo Liu, Chao Zhuo

**Affiliations:** 1State Key Laboratory of Respiratory Disease, The First Affiliated Hospital of Guangzhou Medical University, Guangzhou 510030, China; 2Department of Respiratory Medicine, The First Affiliated Hospital of Guangzhou Medical University, Guangzhou 510030, China; 3Department of Respiratory and Critical Care Medicine, The First Affiliated Hospital of Sun Yat-sen Univesity, Guangzhou 510030, China

**Keywords:** New Delhi metallo-β-lactamase, IncX3 plasmid, fitness cost, adaptive evolution, genetic conflict, carbapenem-resistant *E. coli*

## Abstract

New Delhi metallo-β-lactamase (NDM)-carrying IncX3 plasmids is important in the transmission of carbapenem resistance in *Escherichia coli*. Fitness costs related to plasmid carriage are expected to limit gene exchange; however, the causes of these fitness costs are poorly understood. Compensatory mutations are believed to ameliorate plasmid fitness costs and enable the plasmid’s wide spread, suggesting that such costs are caused by specific plasmid–host genetic conflicts. By combining conjugation tests and experimental evolution with comparative genetic analysis, we showed here that the fitness costs related to *ndm*/IncX3 plasmids in *E. coli* C600 are caused by co-mutations of multiple host chromosomal genes related to sugar metabolism and cell membrane function. Adaptive evolution revealed that mutations in genes associated with oxidative stress, nucleotide and short-chain fatty acid metabolism, and cell membranes ameliorated the costs associated with plasmid carriage. Specific genetic conflicts associated with the *ndm*/IncX3 plasmid in *E. coli* C600 involve metabolism and cell-membrane-related genes, which could be ameliorated by compensatory mutations. Collectively, our findings could explain the wide spread of IncX3 plasmids in bacterial genomes, despite their potential cost.

## 1. Introduction

The emergence and dissemination of carbapenem resistance conferred by New Delhi metallo-β-lactamase (NDM) has become a major concern worldwide [1]. Of the plasmids reported to carry *bla*_ndm_, IncX3-type plasmids are the most common. The *ndm*/IncX3 plasmids have been detected in Asia, Europe, and North America, though strains from China carry the highest proportion of *ndm*/IncX3 (68.4%) [2,3].

Plasmids reduce the competitiveness of the host bacteria and so confer a fitness cost [4], which is considered a major barrier to plasmid maintenance in microbial populations [5]. This cost in fitness has been attributed to plasmid properties, such as the number of copies, size of the plasmid, and related metabolic burden [4,5,6,7]. The *ndm*/IncX3 plasmid was considered to confer a low fitness cost because of its low copy number, minor metabolic burden, and stable genetic backbone [8,9,10,11]; however, high fitness costs have been reported in some *E. coli* strains [12,13]. The mechanism of the diversity of fitness costs while carrying the *ndm*/IncX3 plasmid remains unclear and is potentially unrelated to the general properties of the plasmid.

Genetic conflicts, which emerge from plasmid–chromosome gene interactions as plasmid entry into the host, are proposed by some researchers as vital mechanisms for plasmid-related fitness costs [14,15]. Plasmids are independently replicating genetic elements, which differ from chromosomes. Genetic conflicts arise when plasmids and chromosomes attempt to independently maximize their own fitness [16]. Such genetic conflicts have manifested as mutations, such as missense variant, deletion, and stop-gained, occurring in chromosomes or plasmids. Frequent conflict between plasmid replication and the host chromosome comes at the cost of host survival, such as the fitness cost reported for *Salmonella enterica serovar Typhimurium* ATCC14028 carrying the MDR IncHI2 plasmid pJXP9 manifesting as chromosomal mutations [17]. These genetic conflicts would likely be strain-specific, depending on the genetic background of the strain. Hall et al. [14] proposed that specific genetic conflicts were responsible for the fitness cost reported for *Pseudomonas fluorescens* SBW25 carrying the megaplasmids pQBR57 and pQBR103; the specific genetic conflict occurred as the SOS-response-related gene *PFLU4242* mutation. However, there have been no studies on the relationship between specific genetic conflicts and the fitness cost of carrying *ndm*/IncX3 plasmids. Our work has previously revealed that the *ndm*/IncX3 plasmids are highly conserved [9], and thus we suggest that specific genetic conflicts that may occur in the host chromosome confer a diverse fitness cost.

Following plasmid-related fitness cost emerging, this could have one of two outcomes: either that plasmids will be lost, which is clearly inconsistent with the current epidemic status of plasmids, or that plasmids and host strains can undergo compensatory evolution, which reduces the cost of plasmid carriage and allows for plasmid persistence. Similarly, for the plasmid to persist, the fitness cost imposed by genetic conflicts conferring targeted deletions and mutations should be compensated by adaptive evolution [14,17,18]. Therefore, one may expect coevolution between the host chromosomes and plasmids to correspond with a reduction in the deleterious effects of the plasmid. We suggest that host strains carrying the *ndm*/IncX3 plasmid initially have a fitness cost, which is compensated by adaptive evolution, and thus may lead to diverse levels of fitness cost, but the mechanism of the diverse fitness cost after coevolution remains unknown.

This study investigates the relationship between diverse fitness costs associated with the *ndm*/IncX3 plasmid and specific genetic conflicts, and how adaptive evolution compensates the fitness cost caused by this specific genetic conflict. We evaluated the mechanisms underlying the fitness costs specifically related to the *ndm*/IncX3 plasmid pHN330 in *E. coli* C600 and performed experimental evolution to explore the potential for compensatory evolution and its mechanism. We found that carrying the *ndm*/IncX3 plasmid generated an elevated fitness cost for *E. coli* C600, suggesting the occurrence of a specific genetic conflict. Adaptive evolution can compensate for the specific plasmid–host chromosome conflicts, ameliorating the cost of the C600 carrying the *ndm*/IncX3 plasmid.

## 2. Materials and Methods

### 2.1. Bacterial Strains and Plasmid

The plasmid pHN330 was isolated in our laboratory as reported in our previous study [19]. We performed conjugation experiments and successfully obtained *E. coli* J53 and C600 transformants containing pHN330; Table 1 summarizes the strain information. Susceptibility to antibiotics was assessed using a standard microdilution method according to the CLSI guidelines [20].

### 2.2. Growth Curve Assay

The growth curves of the strains were determined by measuring the optical density (OD) at 600 nm (BioTek, Vermont, USA). Briefly, cultures grown in Luria–Bertani (LB) broth overnight at 37 °C were diluted 1:1000 in fresh LB broth and grown at 37 °C with shaking (200 rpm) for 24 h. The OD_600_ of the culture was measured once every hour for a total of 24 h [21].

### 2.3. Competition Experiments

To assess the fitness impact of pHN330 carriage on the host, transformants and host strains were paired to perform competition experiments. The overnight bacterial suspension of paired strains was adjusted to 0.5 McFarland, mixed at a ~1:1 ratio, and 50 μL of the mixture was seeded into 5 mL of LB broth for 24 h of incubation (37 °C, 200 rpm). At 0 and 24 h, 100 μL of the culture was diluted and the diluted sample was transferred to LB agar or LB agar supplemented with meropenem (1 mg/L). After incubation at 37 °C for 18–24 h, colony forming units (CFUs) were counted and the competitive index (CI) was calculated as previously described [22]: CI = (transconjugant CFU/original CFU)/(inoculated transconjugant CFU/inoculated original CFU).

### 2.4. Experimental Evolution

Briefly, the C600T strain was inoculated into 5 mL of LB broth with or without imipenem (2 mg/L). These cultures were considered generation 0 for each lineage. We transferred 5 μL of the culture daily to 5 mL fresh LB broth with imipenem (2 mg/L) or without antibiotics over 50 d (10 generations per day, 500 generations total). PCR analysis confirmed that all the progeny strains were *ndm*-positive [19].

### 2.5. Whole-Genome Sequencing and Bioinformatic Analysis

Whole-genome sequencing was performed on all strains. Briefly, single colonies were grown for 16 h in LB broth at 37 °C and bacterial DNA was extracted using a bacterial DNA extraction kit (Omega Bio-Tek, USA) according to the manufacturer’s instructions. Whole-genome sequencing was performed by Novegene (Novogene BioTech, Beijing, China) on an Illumina Novaseq platform. Genomes were assembled using Shovill version 1.0.4 [23] and annotated using Prokka software [24]. Single nucleotide polymorphism (SNP) analysis was performed using Snippy [25]. The online tool DAVID [26] was used to annotate SNP genes according to biological function and cellular components via Gene Ontology analysis [27]. The genome of the *E. coli* strains were submitted to GenBank under accession number PRJNA907472.

### 2.6. Statistical Analysis

Data were analyzed using SPSS software version 25.0 (IBM SPSS, Chicago, IL, USA). CIs and growth rates were compared between groups using a paired *t*-test, with statistical significance set at *p* < 0.05.

## 3. Results

### 3.1. Fitness Cost of Carrying IncX3 Plasmid

The pHN330 plasmid harboring *bla*_ndm_ and the studied strains acquired resistance to carbapenems after carrying the IncX3 plasmid. We performed antibiotic susceptibility tests (Table 2) and found that all strains carrying pHN330 were resistant to imipenem and meropenem, with MIC values 32- to 512-fold higher than the control strains.

Competition experiments and growth curves were used to assess the fitness cost of carrying pHN330 (Figure 1). We found that when J53 was the host, there were no significant differences in growth rates between the transformants and control bacteria (*p* > 0.05); when C600 was used as the host, the growth rate of the transformants was lower than that of control bacteria (2.47 vs. 2.16, *p* < 0.0001).

Further competition experiments (Figure 2) showed that the fitness cost of carrying the IncX3 plasmid was minor for the J53 strain (CI: 0.92 ± 0.03) but carrying the pHN330 had a clear fitness cost for the C600 strain (CI: 0.15 ± 0.04).

### 3.2. Comparative Analysis of E. coli C600 and E. coli C600/pHN330

To explore the mechanism conferring the fitness cost related to the IncX3 plasmid, we analyzed whether SNPs occurred in the plasmid or chromosomal genes by comparative genomics. We detected several SNPs in J53 (8) and C600 (29), but none were observed in the IncX3 plasmid (Table 3 and Appendix A).

After J53 obtained the IncX3 plasmid, no SNPs were found in the coding sequence (CDS) regions, and all SNP positions were located in intergenic regions (Table 3). In contrast, after C600 obtained pHN330, 10 non-synonymous SNPs occurred in the CDS region, which were located within the *ptrA*, *lpdA*, *bamA*, *melA*, *gatA*, *mdoB*, *yihV*, *mntH*, *trkA*, and *ydfK* genes, which we termed SNP genes (Table 4).

Functional classification and clustering analyses with DAVID of all SNP genes in the C600 strain (Appendix A) indicated that the *bamA*, *mdoB*, *mntH*, and *trkA* genes were associated with the cell membrane, and *lpdA*, *ptrA*, *gatA*, *yihV*, *ydfK*, and *melA* were related to metabolism.

### 3.3. IncX3 Plasmid Maintenance in E. coli C600 after 500 Passages

Plasmid–host coevolution can gradually reduce the fitness cost of plasmids. To verify whether the same phenomenon exists in C600T, the experimental evolution of C600T was assessed over 50 d without antibiotics or with imipenem 2 mg/L. pHN330 was consistently present in all progeny strains, as confirmed by the PCR assay. The resistance profiles of C600T before and after laboratory evolution (Table 2) showed that the MIC of meropenem was unchanged after passage without antibiotics but decreased 4-fold after passage with imipenem (from 64 mg/L to 16 mg/L), while the MIC of imipenem decreased 2-fold after passage with and without antibiotics (from 64 mg/L to 32 mg/L).

### 3.4. Fitness Cost of Progeny Strains Decreased Compared to C600T Strain

After passages without antibiotics or with imipenem (2 mg/L), the growth rates of C600N500 and C600Y500 significantly increased compared to that of the C600T strain but did not recover to C600 levels. The CIs of C600N500 (0.22 vs. 0.14, *t* = −4.793, *p* = 0.003) and C600Y500 (0.28 vs. 0.14, t = −7.328, *p* = 0.001) were clearly higher than that of C600T (Figure 3).

### 3.5. SNPs Occurring in the CDS Region Reduced the Fitness Cost of C600T

To investigate the mechanism underlying the decreased fitness cost of IncX3 plasmid-carrying strains after evolution, progeny strains and transformants were subjected to comparative genomic analysis (Table 5 and Appendix A). SNPs were present in all progeny strains (C600N500:28; C600Y500:27). After passages without antibiotics or with imipenem (2 mg/L), the progeny strains had more SNPs in the CDS region (C600N500:17; C600Y500:11).

As shown in Table 6, C600N500 had 14 SNP genes, *gudD*, *araC*, *dgt*, *bamA*, *rfaF*, *lptF*, *ycjX*, *bcsG*, *pepQ*, *mshA*, *sfmD*, *paaJ*, *mreC*, and *tfaE*, and C600Y500 had 10 SNP genes, *gloA*, *ydiF*, *fliY*, *hycF*, *caiA*, *bamA*, *rlmB*, *sbp*, *acnA*, and *yghQ*. A reverse mutation in the *bamA* gene was observed in the absence of antibiotic passages and with imipenem passages (1882T > G p.Phe628Val, after passages, reverse).

Further gene functional classification and clustering analyses of the non-synonymous SNP genes were performed in the progeny strains. Among the SNP genes that occurred in C600N500, *bamA*, *bcsG*, *lptF*, *mreC*, *sfmD*, *mshA,* and *rfaF* were related to the cell membrane, and *gudD*, *araC*, *pepQ*, *paaJ*, and *dgt* were related to metabolism (Appendix A). Among the SNP genes found in C600Y500, *fliY*, *bamA*, *yghQ*, and *sbp* were related to the cell membrane, and *caiA*, *gloA*, *acnA*, *rlmB*, *hycF*, and *ydiF* were related to metabolism (Appendix A). The number of SNP genes and gene clusters in the progeny strains and transformants are shown in Figure 4.

## 4. Discussion

The *ndm*/IncX3 plasmid is thought to confer a low fitness cost that contributes to its worldwide occurrence [10,11]. However, some studies have also reported high fitness costs [12,13], though the variety of fitness costs and the mechanism involved remain unclear. Our group has been exploring IncX3 plasmid prevalence for several years [8,9,19]. In the present study, we found that the *ndm*/IncX3 plasmid led to an elevated fitness cost in the C600 strain, for which specific genetic conflicts were responsible and related to SNPs in genes of the cell membrane and sugar metabolism. Adaptive evolution can compensate for specific plasmid–host chromosome conflicts through SNPs in genes related to nucleotide metabolism, strain biofilm formation, LPS synthesis, antioxidant stress, cytoskeleton, and short-chain fatty acid metabolism, ameliorating the cost of the C600 carrying *ndm*/IncX3 plasmid. This indicates that diversity in fitness costs is likely common when evaluating plasmid carriage across hosts, and genetic conflicts may provide relevant insights.

In the present study, plasmid-carrying *E. coli* J53 and C600 showed different fitness costs, which is consist with previous studies that indicated that *ndm*/IncX3 plasmids cause diverse fitness costs [12,13]. Subsequent comparative genomic analysis revealed that the fitness cost of carrying *ndm*/IncX3 plasmids is associated with several genetic mutations. Plasmid-carrying *E. coli* C600 had more SNPs in CDS region genes and fewer in intergenic regions on the chromosome compared to *E. coli* J53, indicating that mutations in intergenic regions incur a low effect on fitness cost. For instance, Buckner et al. [28] showed that the SNPs of *Klebsiella pneumoniae* Ecl8 carrying the pKpQIL plasmid were mainly in the intergenic regions with low fitness cost. Wang et al. [29] found that mutations in *gyrB* resulted in considerable costs, while carrying the *OqxAB*-encoding plasmids pHNGC59 and pHNFD436. Thus, mutations in functional CDS regions play an important role in generating fitness costs.

We analyze what specific genetic conflicts are related to the function of the SNP genes mutated in the CDS region of *E. coli* C600. The functions of the non-synonymous SNP genes in C600 (*ptrA*, *lpdA*, *bamA*, *melA*, *gatA*, *mdoB*, *yihV*, *mntH*, *trkA*, and *ydfK*) were related to the cell membrane and sugar metabolism (Appendix A). Overall, the SNPs occurring in the metabolism-related genes (*lpdA*, *melA*, *gatA*, and *yihV*) reduce the efficiency with which *E. coli* C600 metabolizes different sugars, resulting in slower growth. The *lpdA* gene is a component of the pyruvate dehydrogenase complex, which codes for lipoamide dehydrogenase and has been reported to promote the growth rate of *E. coli* by glucose metabolism [30]. *MelA*, *gatA*, and *yihV*, which are involved in the metabolism of other sugars (galactitol, melibiose, and sulfoquinovose) in addition to glucose, have been reported to restrict bacteria from utilizing essential nutrients and reduce growth rates [31,32,33]. Mutants occurring in the cell-membrane-related genes (*bamA*, *trkA*, and *mdoB*) affected the stability of *E. coli* C600 membranes, reduced resistance to osmotic pressure, and promoted potassium ion imbalance, resulting in the decrease in bacterial growth. The *bamA* gene is an outer membrane protein assembly factor, which encodes a component of the BAM complex that is essential for the construction and maintenance of the outer membrane [34,35]. Phosphoglycerol transferase I—encoded by the *mdoB* gene—is involved in the biosynthesis of membrane-derived oligosaccharides and has been reported to affect the regulation of osmolarity and the stability of the bacterial membrane [36]. *TrkA*, the NAD-binding component of the Trk potassium transporter, is responsible for maintaining potassium ion homeostasis under hypotonic conditions [37]. Consequently, the co-mutations in these metabolic and cell-membrane-related genes reduced the ability of pHN330 carrying *E. coli* C600 to utilize different sugars, affected the membrane stability, decreased the ability to resist various environmental stresses, and caused a potassium ion imbalance, resulting in fitness costs.

Plasmid fitness costs related to bacterial metabolism and cell membrane function have been previously reported. Alvaro et al. [38] found that carrying different plasmids altered the expression of metabolic genes in *Pseudomonas aeruginosa* PAO1, which altered the regulation of energy production, lipid transport, and carbohydrate transport. *E. coli* DH10 carrying the IncA/C2 plasmid pAR060302 has also been reported to have an elevated fitness cost [39], specifically through the upregulated expression of the plasma membrane and active transmembrane transport genes [40].

In nature, the cost of carrying plasmids can be compensated through adaptive evolution [41]. In the present study, we found that the cost could be compensated by co-evolution, but the underlying mechanisms of host–plasmid coevolution compensating for the fitness cost are still unknown. Comparative genomic analysis between progeny strains and transformants revealed that *bamA* showed a reverse mutation in the compensated strains C600N500 and C600Y500. BamA is an essential component of the Omp85 protein superfamily and is well conserved in all Gram-negative bacteria [34,42]. BamA is important for the activity of the conserved β-barrel assembly machinery, which is responsible for the assembly of the outer membrane β-barrel conformation in Gram-negative bacteria [43,44]. BamA, the component of β-barrel membrane proteins, is proven to be vital for nutrient import, signaling, motility, and survival of Gram-negative bacteria [45]. Wu et al. found that the BamA protein depleted in *E coli* JCM166 results in cell death [46]. Thus, our results are consistent with the function of the *bamA* gene; mutants that occurred in the *bamA* gene resulted in a decreased growth rate and, after compensation, the *bamA* gene reverse mutant increased the growth rate of the C600N500 and C600Y500 strains. Future studies are needed to explore the role of the *bamA* gene in determining the fitness cost of another IncX3 plasmid bearing *E. coli*.

In addition to the *bamA* gene, GO clustering analysis suggested that other non-synonymous SNP genes occurred in progeny strains C600N500 and C600Y500, which were also related to the cell membrane, nucleotide, and short-chain fatty acid metabolism (Appendix A). The metabolism-related non-synonymous SNP gene *dgt* occurred in C600N500 and is associated with nucleotide metabolism. The *dgt* gene encodes for dGTPase, responsible for the hydrolysis of dGTP to deoxyguanosine and triphosphate. Mutants in the *dgt* gene can cause mismatches during DNA replication [47]. The cell-membrane-related non-synonymous SNP genes *bcsG*, *lptF*, *rfaF*, and *mreC* occurred in C600N500 and are associated with biofilm formation, LPS synthesis, and cell cytoskeletal formation. The gene *bcsG*, which is involved in the production of exopolysaccharide cellulose, has been reported to affect biofilm formation in *E. coli* [48]. The *lptF* and *rfaF* genes are essential for lipopolysaccharide formation, which is critical for cell motility, intestinal colonization, and biofilm formation, and encode for major extracellular polymers [49,50,51]. The gene *mreC* encodes for cytoskeletal proteins that regulate cell morphology and survival and is highly conserved in bacteria [52].

Adaptive evolution can compensate for fitness costs through altered metabolism and cell membrane function. Elias et al. found that the fitness cost of *K. pneumoniae* after obtaining ceftazidime-avibactam resistance can be compensated through adaptive evolution through the downregulated expression of phospholipids and LPS biogenesis and modification [53]. Previous studies have found that *P. fluorescens* carries pQBR103 at an elevated cost and can be compensated by a mutant in the *gacA/gacS* two-component system [54], which regulates energy metabolism and central intermediary metabolism [55]. Ma et al. found that after evolution, mutant DNA replication in the *dnaK* gene compensated for the fitness cost of *E. coli* carrying the *mcr*/IncHI plasmid [56].

Oxidative stress resistance and the metabolism of short-chain fatty acids were related to the adaptive evolution of C600Y500 under antibiotics. The *ydiF* gene encodes acetate-CoA transferase, which is a short-chain acyl-CoA involved in the metabolism of short-chain fatty acids [57]. FliY is an *E. coli* L-cysteine/L-cystine importer that cooperates with the L-cysteine exporter EAMA in the L-cysteine/L-cystine shuttle system involved in oxidative stress resistance [58] and has been reported to affect the growth of *E. coli* in LB broth [59]. The *gloA* gene encodes glyoxalase I involved in the glutathione (GSH)-dependent glyoxalase I pathway [60], which influences the metabolism of methylglyoxal; a high concentration of methylglyoxal has been reported to lead to bacterial growth stasis and even death [61]. The *fliY*, *gloA*, and *acnA* gene mutations in C600Y500, which are related to different pathways involved in oxidative stress resistance, may be related to the selection pressure of imipenem on C600Y500. Wang et al. found that imipenem selection pressure in *Acinetobacter baumannii* stimulated the strain to produce hydroxyl radicals [62]. Another study found that ampicillin induced the formation of hydroxyl radicals through oxidative damage, which led to bacterial death [63].

Our results suggest that while IncX3 plasmids are already known to have a low fitness cost, occasional elevated fitness costs can occur. This suggests that IncX3 plasmids could have initially been high-fitness-cost plasmids; however, through host–plasmid coevolution, the IncX3 plasmid became a low-fitness-cost plasmid. Our findings indicate that the host–plasmid adaptive coevolution process is a trade-off between plasmid and host chromosome fitness, in which the IncX3 plasmid is an approximate perfect plasmid, and thus the plasmid and host reach a compromise manifested as host chromosome mutations alleviating the plasmid-borne fitness cost. *S. Typhimurium* ATCC 14028 and IncHI2 plasmid pJXP9 plasmid evolutionary analysis also confirmed the plasmid–host trade-off [17]. This study regarding the mechanism of the fitness cost while carrying the *ndm*/IncX3 plasmid provides insight about how bacteria maintenance plasmids. Our findings provide a reasonable representation of the evolution of the IncX3 plasmid and provide a better understanding of its prevalence.

The present study has some limitations. We found that *bamA* may be a vital factor compensating for the fitness cost related to specific genetic conflicts. The role of BamA mutations should be confirmed in complementation experiments. Fitness costs and compensatory mechanisms are related to diverse metabolic pathways; thus, in-depth metabolomic and transcriptomic analyses are needed to provide more specific insights across bacterial strains and species.

In conclusion, we report that the fitness cost of the *ndm*/IncX3 plasmid was related to specific genetic conflicts that were ameliorated by compensatory mutations following coevolution, specifically through the *bamA* gene reverse mutation. Our study provides a clearer understanding of the mechanisms related to the fitness cost of carrying plasmids and their adaptive evolution with host strains, which is essential for predicting the prevalence and evolution of plasmid-mediated horizontal gene transfer.

## Figures and Tables

**Figure 1 pathogens-12-00137-f001:**
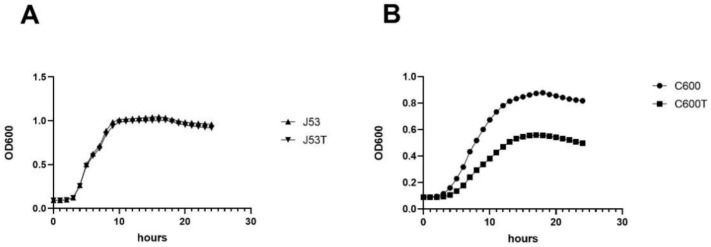
Growth curves of host strains and transformants. (**A**) Comparison of the growth curves for J53 and J53T showed no significant changes. (**B**) Comparison of growth curves for C600 and C600T; the growth rate was significantly decreased in C600T.

**Figure 2 pathogens-12-00137-f002:**
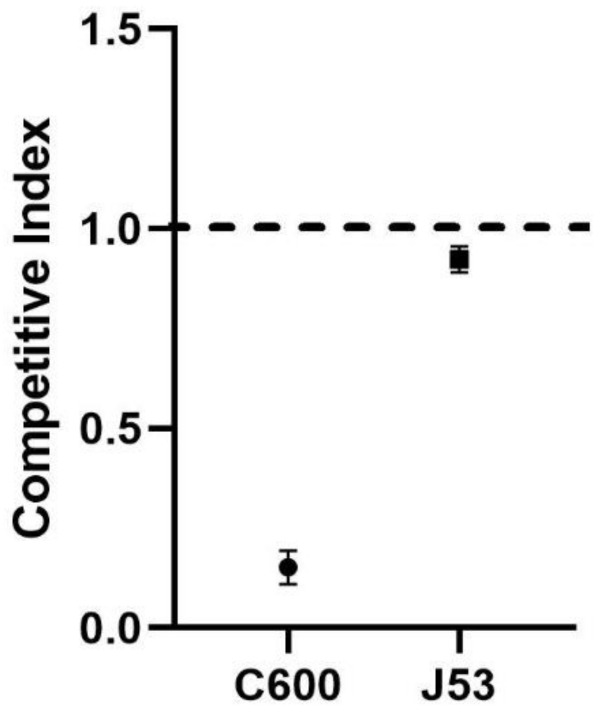
The competitive indices of the C600 and J53 strains. The transformants and host strains were paired to perform competition experiments and the competitive index (CI) was calculated. CI values < 1 indicate that the carried plasmid generates a fitness cost, with smaller CI values indicating higher fitness costs.

**Figure 3 pathogens-12-00137-f003:**
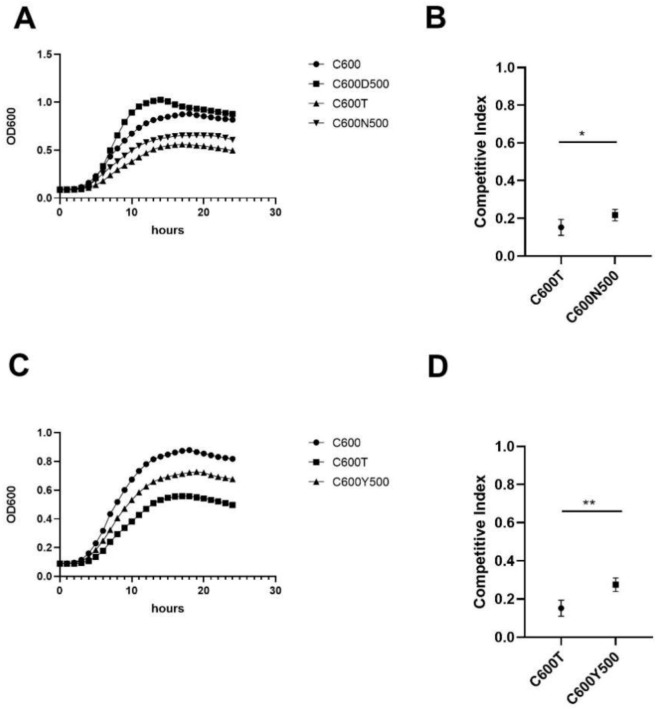
Experimental evolution of C600T without antibiotics or with imipenem (2 mg/L). (**A**,**C**) Comparison of the growth curves for C600N500, C600Y500, and C600T. (**B**,**D**) After evolution without or with antibiotics, the competitive index of C600N500 and C600Y500 increased significantly compared to that of C600T. * *p* = 0.003; ** *p* = 0.001.

**Figure 4 pathogens-12-00137-f004:**
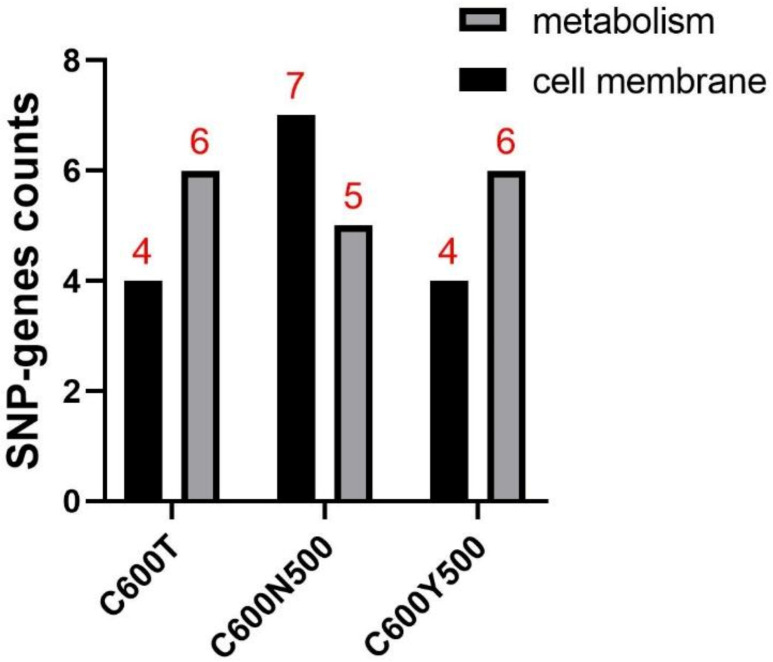
SNP gene function classification. SNP: single nucleotide polymorphism. GO clustering analysis was performed on non-synonymous SNP genes, which could be divided into metabolism- and cell-membrane-related genes. Numbers above the bars represent the number of non-synonymous SNP genes.

**Table 1 pathogens-12-00137-t001:** Strain descriptions.

Strain	Plasmid Replicon	Description
C600	None	model *E. coli* strain
C600T	IncX3	C600 transformant
C600N500	IncX3	500th passage of C600T without antibiotics
C600Y500	IncX3	500th passage of C600T with 2 mg/L imipenem
C600D500	None	500th passage of C600
J53	None	model *E. coli* strain
J53T	IncX3	J53 transformant

**Table 2 pathogens-12-00137-t002:** Resistance profiles of the host strains and transformants.

Strain	MIC (mg/L)						
	MEM	IPM	FEP	CAZ	ATM	AMK	LEV
C600	0.125	0.25	0.125	1	0.25	0.125	0.125
C600T	64 (512-fold) ^a^	64 (256-fold) ^a^	32 (256-fold) ^a^	128 (128-fold) ^a^	8 (32-fold) ^a^	0.125	0.125
C600N500	32	32	32	128	8	0.125	0.125
C600Y500	16	32	32	128	8	0.125	0.125
J53	0.125	0.125	0.125	0.125	0.5	4	0.125
J53T	4 (32-fold) ^a^	16 (128-fold) ^a^	64 (512-fold) ^a^	64 (512-fold) ^a^	128 (512-fold) ^a^	4	0.125

MEM—meropenem, IPM—imipenem, FEP—cefepime, CAZ—ceftazidime, ATM—aztreonam, AMK– amikacin, LEV—Levofloxacin. ^a^ Fold increase in MIC values of the transformants compared to the model strain.

**Table 3 pathogens-12-00137-t003:** Number of SNPs in the host strains carrying the pHN330 plasmid.

Strain		SNP Counts	
	Total SNPs	CDS Count	Intergenic Sequence Count
J53T	8	0	8
C600T	29	11	18

SNP—single nucleotide polymorphism, CDS—coding sequence. SNPs differed between J53 and C600, with J53 having no SNP in the CDS region and C600 having 11 SNPs in the CDS region. SNP gene—gene occurring in the CDS region.

**Table 4 pathogens-12-00137-t004:** Non-synonymous SNP genes in host strains carrying the pHN330 plasmid.

Strain	Non-Synonymous SNP Gene
J53T	None
C600T	*ptrA*, *lpdA*, *bamA*, *melA*, *gatA*, *mdoB*, *yihV*, *mntH*, *trkA*, and *ydfK*

**Table 5 pathogens-12-00137-t005:** SNP counts in progeny strains.

Strain		SNP Counts	
	Total SNPs	CDS Count	Intergenic Sequence Count
C600N500	28	17	11
C600Y500	27	11	16

SNP: single nucleotide polymorphism, CDS: coding sequence. After adaptive evolution, the SNP was located in the CDS region, and the intergenic sequence region occurred in the progeny strains.

**Table 6 pathogens-12-00137-t006:** Non-synonymous SNP genes in progeny strains.

Strain	Non-Synonymous SNP Gene
C600N500	*gudD*, *araC*, *dgt*, *bamA*, *rfaF*, *lptF*, *ycjX*, *bcsG*, *pepQ*, *mshA*, *sfmD*, *paaJ*, *mreC*, and *tfaE*
C600Y500	*gloA*, *ydiF*, *fliY*, *hycF*, *caiA*, *bamA*, *rlmB*, *sbp*, *acnA*, and *yghQ*

SNP gene—gene occurring in the CDS region.

## Data Availability

The original contributions presented in the study are included in the article/Supplementary Material, and further inquiries can be directed to the corresponding authors.

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
