# Peer review of "Adaptive Evolution Compensated for the Plasmid Fitness Costs Brought by Specific Genetic Conflicts"

_pathogens, 2023, doi:10.3390/pathogens12010137_

Round 1

Reviewer 1 Report

ADAPTIVE EVOLUTION OF THE PLASMID FITNESS COSTS BROUGHT BY SPECIFIC GENETIC CONFLICTS 

The study aimed to analyze plasmid fitness costs in NDM producing E. coli harbouring X3 plasmid. It found that the fitness cost of the plasmid carriage was related to the mutations of the chromosomal genes important for the suger metabolism and cell membrane functions. The topic is original and up to data. The bibliographical references on this issue are rare. The study clarified the reasons for the spread of X3 plasmid.

The aim of the study is novel, clear and well defined. The methods are su accurate and reproducible. The results are well represented and sound. The discussion and conclusions are balanced and adequately suppported by the data presented in the result section. The references are up to date and the quality of English language is excellent. The title and the abstract convey with what was found. The manuscript adhere with the relevant standards for reporting of the data.

MINOR COMMENTS

1.      What was the rationale for choosing X3 plasmid? There are also other plasmids carrying blaNDM genes for instance L/M which are also widespread.

2.      What was the reason for choosing E. coli? NDM carbapenemases are more frequent in K. pneumoniae.

3.      The aim of the study should be better explained in the abstract.

Reviewer 2 Report

The ndm/IncX3 plasmids imposed fitness costs on E. coli C600. Li et al. showed that the costs were caused by mutations of multiple host chromosomal genes related to sugar metabolism and cell membrane function. During adaptive evolution, the costs were ameliorated by mutations in genes associated with oxidative stress, nucleotide and short-chain fatty acid metabolism, and cell membrane. The manuscript is overall in its good shape. 

Major comments: 

1. I would like the authors to dig deeper into the fitness cost imposed by plasmid carriage. Bergstrom (doi.org/10.1093/genetics/155.4.1505) first proposed the idea that there are only two options for a plasmid because of the cost—1. get cured, 2. be integrated into a chromosome. I am glad you cited the San Millan paper. Another paper (doi.org/10.1093/gbe/evz197) also showed that plasmid mutations can lead to higher resistance under Tc pressure by increasing plasmid copy number; however, because of the higher burden imposed by a higher copy number, these mutations were eventually purged from the population. 

Minor comments: 

1. Line 34, "blandm" → "blandm"

2. Line 110, "2 g/L" or "2 mg/L" (= 2 µg/mL)? I am not sure if the unit is correct in terms of the AB working concentration. I have the same question for Table 2.
